# Ball Milling Medium May Tune the Self-Curing Property and Root Canal Microleakage of β-Dicalcium Silicate-Based Cement

**DOI:** 10.3390/ma15145043

**Published:** 2022-07-20

**Authors:** Youyang Zheng, Xianyan Yang, Shuxin Liu, Yuyue Xu, Siqi Bao, Yunyi Wang, Yuhan Liu, Feng Zhang, Zhongru Gou

**Affiliations:** 1Department of Stomatology, The Second Affiliated Hospital, School of Medicine Zhejiang University, Hangzhou 310009, China; zr_zyy@zju.edu.cn; 2Bio-Nanomaterials and Regenerative Medicine Research Division, Zhejiang-California International Nanosystems Institute, Zhejiang University, Hangzhou 310058, China; xyyang2008@zju.edu.cn; 3School of Stomatology, Zhejiang University School of Medicine, Hangzhou 310006, China; 3180102450@zju.edu.cn (S.L.); 3180102481@zju.edu.cn (Y.X.); 3180102440@zju.edu.cn (S.B.); 4Department of Stomatology, The Children’s Hospital, Zhejiang University School of Medicine, National Clinical Research Center for Child Health, Hangzhou 310003, China; 22118765@zju.edu.cn; 5School of Stomatology, Zhejiang Chinese Medical University, Hangzhou 310053, China; yuhanliu0419@163.com

**Keywords:** ball milling medium, attapulgite, anti-microleakage, β-dicalcium silicate, root canal sealer

## Abstract

It is still a challenge to overcome the extended setting process of pure Ca-silicate as root canal fillers. We investigated the effects of attapulgite (a basic hydrous silicate of magnesium and aluminum) and ball-milling liquid medium on the self-curing properties of conventional β-dicalcium silicate (C2Si)-based cements. It was shown that a minor amount of attapulgite nanofibers (1–4%) had only a slight influence on setting time but caused a large increase in compressive resistance and structural stability. In particular, the ball milling media with different acetone/water ratios (3:0, 2:1, 1:2, 0:3) could directly influence the particle size distribution of C2Si powders, and the co-existence of liquid media (2:1 or 1:2) may be beneficial for shortening the setting time, enhancing early-stage compressive strength, and significantly improving the anti-microleakage ability of cement. Moreover, the composite cements also exhibited appreciable antibacterial efficacy in vitro. These findings demonstrated that the physicochemical properties of the Ca-silicate powders could be tuned by adding a minor amount of inorganic silicate nanofibers and a simple ball milling condition, and such a facile strategy is favorable for developing novel (pre-mixed) Ca silicate-based cements as root canal sealers.

## 1. Introduction

Infection-derived endodontic treatment is mainly dependent on thorough microbial elimination, intermediate complete sealing of the root canal, root canal shape, and even fast apatite re-mineralization in the dentinal tubules around the root canal wall [1,2,3]. Generally, a root canal filler is be expected to have excellent integration between the sealer and dentinal wall and should also show good chemical stability and apatite-inducing deposition potential to always close any microscopic cavities or tubules. Moreover, the sealer is expected to have appropriate self-curing efficacy to facilitate sealing and to avoid microleakage. In this respect, organic sealers such as epoxy or methacrylate resin are suboptimal for clinical usage [4].

In the past two decades, some attention has been paid to Ca-silicate-based cements used for endodontic treatment because of their appreciable apatite mineralization ability and good biocompatibility [5,6,7,8]. It has been claimed that these inorganic materials, such as the β phase of dicalcium silicate (β-Ca_2_SiO_3_; C2Si), may release calcium ions in aqueous medium, and the silica surface may readily induce bone-like apatite deposition [5,6]. In fact, mineral trioxide aggregate (MTA) and iRoot, two types of mixtures of Ca-silicate/-phosphate composites, have been available for root canal sealing, apexification, and root-perforation repair in endodontics [8]. Unfortunately, a few studies show that the potential of discoloration or extended setting time for these sealer have stimulated the development of alternative filling materials [9,10,11].

C2Si, as a main component in Portland cement, has been widely studied as a promising bone defect implant; furthermore, this Ca-silicate has attracted much interest in endodontic therapy because of its good self-curing and antibacterial potential compared to some organic sealers [8]. C2Si particles can react with water to transform to calcium-silicate-hydrate gel (C-S-H) and then develop appreciable mechanical strength with a prolonged self-curing process time. Although our previous studies have demonstrated that a bone mineral-like apatite layer may deposit on the surface of C2Si cement within 24 h, and the osteoblasts may readily spread on the cement after a 24-h culture [12], the pure C2Si cement showed a slow curing reaction, and the addition of gypsum is detrimental to the chemical stability due to fast bio-dissolution in aqueous medium. Thus, the slow curing nature of C2Si is disadvantageous for its clinical operation in endodontic treatment.

In order to accelerate the self-curing process, some attempts have been made to improve the properties of silicate-based cements. For instance, Wang et al. added CaCl_2_ into the tricalcium silicate (Ca_3_SiO_4_; C3Si) cement to accelerate its curing and hardening ability [13,14]. Zhou et al. found that Mg-phosphate cement could act as curing accelerator of the C3Si-based premixed cements [7]. Moreover, some previous studies have found that the introduction of gypsum or calcium carbonate was able to accelerate the self-curing and mechanical resistance of the Ca-silicate-based paste [15,16,17,18]. Moreover, we have also confirmed that zinc or copper doping may enhance the antibacterial potential and osteoblast activity of C2Si cement [5]. However, the slow hydration nature of C2Si cement has not been resolved thoroughly, though secondary phasic additives have been widely used to accelerate the curing process of the pastes.

On the other hand, attapulgite (ATT), a clay with a fiber-like morphology, belongs to the family of hydrous Mg–Al–silicate minerals. ATT nanofibers have attracted increased attention as a sorbent [19], polymer additive [20], and synthesis additive for new materials [21] due to its structural morphology and surface properties. The abundant hydroxyl groups on ATT’s surface make it a good binder as well [22]. These hydroxyl groups are dehydrated and cross-linked between adjacent binder particles after calcination, thereby increasing the mechanical strength of the material substrate. In this respect, ATT is a valuable candidate for potentially improving the curing properties of C2Si in endodontic treatment and may be favorable for accomplishing long-term sealing in root canal systems.

Based on these concerns, we explored the effect of pre-curing treatment by the addition of ATT and the ball milling technique with different aqueous media on the physicochemical and curing performances of C2Si-based cements, with the aim of testing their potential as root canal sealers. Particularly, their initial and final setting time, compressive strength, anti-microleakage potential, and apatite re-mineralization in vitro were compared with each other. The experimental results demonstrated that the ball milling medium could readily tune the curing property and helped to produce a significantly shortened setting time, appreciable compressive resistance, and the expected anti-microleakage functions.

## 2. Materials and Methods

### 2.1. Materials

Analytic reagent-grade trishydroxymethyl aminomethane (Tris), colloidal SiO_2_, ammonia (NH_3_·H_2_O, ~28 wt%), calcium nitrate (Ca(NO_3_)_2_·4H_2_O), acetone, and absolute ethanol were purchased from Sinopharm Chemical Reagent Co., Shanghai, China. Artificial saliva (ISO/TR10271) and ATT were bought from Shanghai Yuanye Bio-Technology Co., Shanghai, China, and Mingguang Mingjiu S&T Co., Shanghai, China, respectively.

### 2.2. C2Si Powder Preparation and Primary Characterization

Ca(NO_3_)_2_·4H_2_O and colloidal SiO_2_ were used directly for preparing the C2Si powders by a sol–gel process [12]. Briefly, the Ca(NO_3_)_2_∙4H_2_O and colloidal SiO_2_ with a nominal CaO/SiO_2_ molar ratio of 1.9:1 was added to the ethanol–water mixture by magnetic stirring. Then, the sol was aged at 60 °C for 72 h, dried at 120 °C for 24 h, and calcined at 850 °C for 3 h. Finally, the calcined powder (40 g; ×4) was planetary ball-milled for 6 h (360 RPM; ~22 °C) in the mixture solution (90 mL) comprised of acetone and deionized water (deH_2_O) with different volume ratios of 3:0, 2:1, 1:2, and 0:3, respectively. The ball mass-to-powder ratio was 2:1, and the diameter of ZrO_2_ balls were 2.4 mm and 7.2 mm (1:1 in mass), respectively. Additionally, the conventional planetary ball-milling treatment (40 g) was also carried out in absolute ethanol medium with a similar solid/liquid ratio, while other conditions remained the same. The ball-milled suspensions were then filtrated and dried at 80 °C overnight.

The phase composition of as-calcined powders was characterized by X-ray diffraction (XRD, Rigaku; Tokyo, Japan). The particle size distribution was determined by laser granulometry on a netasizer nano (Malvern, S90, Malvern, UK). The thermogravimetric and differential thermal analyses (TG/DTA) for the as-dried powders were carried out on a TG/DTA6200 from TA Instruments with a 10 °C·min^−1^ heating rate under an air atmosphere. The dried powders were also observed by scanning electron microscopy (SEM; JEM-6700F; Tokyo, Japan).

### 2.3. Setting Time Measurement

The pastes were prepared by mixing the C2Si/ATT*x* composite powders (ATT containing (*x*): 0%, 1%, 2%, 4%, 16%, 24%, and 32%) with 2.0 wt% CaCl_2_ aqueous solution in a series of liquid-to-solid (L/S) ratios of 0.7, 0.8, and 0.9 mL·g^−1^, respectively. The initial (*I*) and final (*F*) setting times were measured with a Vicat needle according to ISO9597-1989E, as reported previously [12]. Briefly, the experiments were carried out on the fresh pastes, which were prepared by mixing 5.0 g of powders with different L/S ratios. Each paste was immediately cast in a stainless steel mold with a diameter of 15 mm and a height of 20 mm, and they were then cured in a water bath in controlled temperature and humidity conditions (37 °C; RH > 95%). To find the two parameters (*I* and *F*), the samples were removed from the incubation chamber and measured using a Vicat needle penetrometer every 2 min. The times from the inorganic pastes being cured at 37 °C to the setting point were used as the setting times. Each group was repeated six times, and the average value was calculated. The results were expressed as mean ± standard difference (SD).

### 2.4. Mechanical Analysis

The pastes were prepared by mixing the C2Si/ATT*x* composite powders with 2.0 wt% CaCl_2_ solution, as mentioned above. Then, the pastes were transferred to Teflon dies (Ø6 × 6 mm) and stored in a 37 °C, 100% humidity condition for different time stages (1–28 days). The mechanical test was conducted on a universal testing machine (Shanghai Zhuoji Co., Shanghai, China) until failure at a loading rate of 1 mm·min^−1^ to obtain compressive strength. Four samples were carried out for each group, and the results were expressed as mean ± standard deviation (SD).

### 2.5. In Vitro Bio-Dissolution and Surface Microstructure Analysis

As for the bio-dissolution (mass loss) test, the paste samples (Ø 6 mm) were prepared with 4.0 g powder (M_0_) and then stored in 100% humidity at 37 °C for 2 and 4 weeks, respectively. The degradation behavior was estimated in 0.05 mM Tris buffer (pH ~7.40; 37 °C) with a cement surface-area to solution volume ratio of 0.1 cm^−1^. Ten percent of the aqueous buffer was refreshed every 24 h. Then, the samples were gently rinsed with absolute ethanol and dried (85 °C) up to mass constancy before weighing (M_t_). The mass decrease (M, %) was calculated according to the formula: M = (M_0_ − M_t_)/M_0_ × 100%. Moreover, the mechanical decay of cements was also determined by compressive tests. Moreover, the surface microstructure of the immersed C2Si/ATT2 samples in artificial saliva (pH ~7.0) at 37 °C for 0–10 days was also determined by SEM observation.

### 2.6. Endodontic Space Sealing Ex Vivo

The single-rooted crack-free human teeth (*n* = 50) with mature apices and single canal configurations were used and pretreated as described in our previous study [6]. The extracted teeth were obtained from the Second Affiliated Hospital, School of Medicine of Zhejiang University, according to standard guidelines approved by the Zhejiang University Ethics Committee (I2022521). First, periodontal ligament tissue was scraped with a scalpel blade, and the crowns of the teeth were removed at the cement to enamel junction. Each root canal was drilled to the apical foramen with sequential files beginning with a size 15 k-file, and then canals were mechanically prepared in a crown down technique with a series of ProFile files (Dentsply Maillefer, Tulsa, OK, USA). Finally, 0.9% NaCl and 5.2% NaOCl were used as irrigation solutions between each file size. The specimens among the 40 pretreated teeth were delivered into the canals using a lentulo spiral.

The teeth (*n* = 40) were divided into four groups of 10 canals, each group being covered with a nail varnish layer, leaving only the apex (~0.8 mm) free for penetration in order to assess the degree of infiltration in the coronal–apical direction and filled with one of the pastes, or used as positive control group. For the positive control, no paste was used, and the surface was coated using nail varnish. All groups were kept at 37 °C, 100% humidity for 24 h, and then they were stored in air at room temperature for 12 h, and finally, they were immersed in 0.5% rhodamine B at 37 °C for 3 days after, under vacuum for the first 20 min.

To measure the length of dye penetration, cross-sections were made longitudinally with a low speed carborundum disk, and digital images of specimen cross-sections were photographed under a stereomicroscope (Olympus SZX7; Tokyo, Japan)-equipped ToupCam^TM^ TP705000A industrial digital camera (Shanghai, China), and then the maximum lengths of penetrations that occurred between the filing materials and the dentinal walls were measured in the longitudinal linear direction by the image processing program ImageJ (NIH, Bethesda, MD, USA). The average value of the maximal depth of dye penetration was calculated.

### 2.7. In Vitro Apatite Re-Mineralization Activity Analysis

The artificial saliva (NaCl, 0.400 g∙L^−1^; KCl, 0.400 g∙L^−1^; CaCl_2_·6H_2_O, 0.795 g∙L^−1^; NaS·2H_2_O, 0.005 g∙L^−1^; Na_2_H_2_PO_4_·2H_2_O, 0.780 g∙L^−1^; urea, 1.000 g∙L^−1^) buffered at pH = ~7.0 was used for in vitro apatite re-mineralization evaluation. Firstly, the teeth were cut to expose the apical root canal. Then, the fresh paste was injected into the 6 mm-long root apex and stored in 100% humidity for 24 h and then immersed in artificial saliva at 37 °C for 72 h. Ten percent of the aqueous buffer was refreshed with fresh artificial saliva every 8 h. The specimen was gently rinsed with ethanol and dried in vacuum overnight. After that, the teeth-filled cements were cut, and the fracture surfaces underwent SEM observation and energy dispersive X-ray spectrometry (EDX) measurements.

### 2.8. Antibacterial Assay for the Immersion Solution of Cements

The fresh C2Si/ATT2 cement was injected into the cylindrical mold to form Ø 18 × 4 mm discs, and the specimens were stored at 37 °C, 100% humidity, for 24 h. Then, the cement was crushed and added into 20 mL of artificial saliva medium for 4 h. *Staphylococcus aureus* (*S. aureus*; ATCC25923) and *Pseudomonas aeruginosa* (*P. aeruginosa*; ATCC27853) originating from American Type Culture Collection (ATCC; Manassas, VA, USA) were employed in this study. These strains were cultured overnight to the logarithmic growth stage, and they were adjusted to about 0.5 MCF (McFarland standard) using sterile Mueller–Hinton Broth (MHB).

Firstly, the different cement extracts with serial dilution (1, 1/2, 1/4, 1/8, 1/16, 1/32) were determined by the microdilution broth method, as described by the National Committee for Clinical Laboratory Standards. The inoculum was adjusted from an exponential-phase culture to yield an initial concentration of approximately 5 × 10^5^ colony-forming units (CFU/mL). The negative control (NC) test tubes received no bacterial suspension. Another test tube (positive control, PC) contained a bacterial suspension without any medication to show the capability of bacterial growth and the production of complete haziness. After 8 h of incubation at 37 °C, the antimicrobial potential was determined in terms of preventing visible growth. Similar to the experimental procedures above, a 10 μL culture of each clear test tube in the bactericidal circle test was inoculated on nutrient agar and incubated at 37 °C for 8 h. After 8 h of incubation at 37 °C, the discs were observed by microscopic observation (Olympus).

### 2.9. Statistical Analysis

Experimental results were expressed as means ± SD. Statistical analysis was carried out using one-way ANOVA, and a *p*-value of less than 0.05 was considered statistically significant.

## 3. Results

### 3.1. Primary Evaluation of the Powders

The C2Si and ATT powders were primarily characterized by XRD analysis (Figure 1). In comparison with the standard XRD pattern of the β phase of C2Si (JCPDS 33-0302), the XRD pattern confirmed that the free CaO was present in the ball-milled powders, even though the main patterns were attributed to β-C2Si. Based on quantitative analysis, it was confirmed that the free CaO was nearly 4.9 wt%. The XRD analysis also indicated that the ATT was highly crystalline powder, free of any other secondary phase. Moreover, the SEM observation indicated that the C2Si particles were irregular aggregates, but the ATT exhibited nanofibrous morphology of two microns in length.

### 3.2. Setting Property Evaluation of C2Si/ATTx

Our primary tests were performed to evaluate the influence of ATT on the initial/final setting time of the C2Si pastes at the L/S ratio of 0.7. Figure 2A shows that the increase of ATT content from 0% to 32% inevitably led to an increase of initial/final setting times of the C2Si-based pastes. Interestingly, the low ATT content (from 1% to 4%) showed a very limited increase of I (24–29 min) and F (55–68 min) values, whereas the high ATT content led to significantly high I (43–50 min) and F (110–140 min) values. Moreover, the compressive strength of the self-cured cements also showed some degree of variation with increasing ATT content (Figure 2B). It is particularly interesting that the low ATT content was specifically favorable for enhancing the compressive resistance with time. The C2Si/ATT2 cement showed 2- to 3-fold higher compressive strength in comparison with the pure C2Si samples (i.e., C2Si/ATT0). Unfortunately, the C2Si/ATT32 samples had lower compressive strength than the pure C2Si.

As for the C2Si/ATT*x* cements, Figure 2C shows that the mass was increased for the C2Si/ATT*x* with low ATT content (≤4%), but high ATT content (16–32%) led to fast mass decay over time in the early stage (~4 weeks), consistent with the increase of ATT content. On the other hand, the mechanical strength exhibited another change with time during immersion in Tris buffer (Figure 2D). The compressive strength was increased within the initial 2 weeks and then decreased from 2 to 4 weeks. Indeed, the C2Si/ATT2 and C2Si/ATT4 samples showed much higher strength than the other samples. However, the C2Si/ATT24 and C2Si/ATT32 showed lower strength than the pure C2Si cement at each time point, implying the high ATT content is unbeneficial for structural densification of the self-cured composite cements.

### 3.3. Systematic Characterization of C2Si after Acetone/Water Ball Milling

The mixtures with different acetone/water ratios (3:0, 2:1, 1:2, 0:3) were used as the ball milling media for as-calcined C2Si powders. It can be observed from Figure 3A,B that the particle size of milled C2Si powder was increased (D_50_, from ~6 μm to 12–13 μm) with increasing water content in the ball milling medium. The ACE0/WAT3 samples showed significantly higher agglomeration than the ACE3/WAT0 (Figure 3A). The average sizes of ACE3/WAT0 and ACE0/WAT3 samples were below 7 μm and nearly 13 μm, respectively (Figure 3B). Moreover, XRD analysis could determine some peaks of Ca-silicate-hydrate (C-S-H) and Ca(OH)_2_ with increasing water content in the medium (Figure 3C). Moreover, the TG-DTA analysis also indicated that the ACE0/WAT3 sample had the highest mass loss (~10%) in comparison with the others (~8%) treated in acetone-containing medium (Figure 3D). This difference may be attributed to one reason, namely, that the water content directly influenced the hydration degree of the C2Si powders, and thus the high crystal water content in the C-S-H aggregates inevitably led to high mass loss in the ACE0/WAT3 sample during TG measurement.

### 3.4. Effect of Ball Milling Condition on Setting Behavior of C2Si-Based Pastes

In order to evaluate the ball milling condition on the influence of setting time, the four groups of C2Si/ATT2 (denoted as ACE3/WAT0, ACE2/WAT1, ACE1/WAT2, ACE0/WAT3) were systematically studied on the L/S ratio from 0.7 to 0.9. As seen in Figure 4A,B, the ACE2/WAT1 and ACE1/WAT2 samples had lower initial and final setting time than the other samples. Moreover, the initial/final setting times exhibited an appreciable increase with the increase of the L/S ratio. It is interesting that the C2Si powders milled in pure water medium (acetone/water = 0:3) indicated much long initial/final setting time (>150 min and 300 min; *p* < 0.01) at the L/S ratio of 0.8 and 0.9. The effects of the C2Si ball milling condition and the L/S ratio on compressive resistance of the C2Si/ATT2 cements with time are shown in Figure 4C,D, which indicated that the compressive strength of cements was increased with prolonging time, but the L/S ratio could significantly influence this mechanical parameter in the early stage. The strength of the samples with an L/S ratio of 0.7 was nearly two-fold higher than that with an L/S ratio of 0.9. However, it is interesting that only the powders milled in pure water (acetone/water = 0:3) showed very low strength during the early stage (~1 day), beside for the situation at the L/S ratio of 0.9.

### 3.5. In Vitro Apatite Re-Mineralization Potential of C2Si-Based Pastes

It is known that Portland cement can maintain stability under natural conditions unless it suffers strong acid erosion. The poly-condensation of silica networks in C2Si hydrates has been widely demonstrated to be biologically stable in humid environments. In this study, the microstructural stability of C2Si/ATT2 composite cements was also confirmed by SEM observation, as shown in Figure 5. It was evident that the ACE3/WAT0 and ACE2/WAT1 samples showed denser surface microstructures than the ACE1/WAT2 and ACE0/WAT3. Moreover, the surface microstructure of each cement group was also maintained stably with the prolongation of immersion time in artificial saliva. In fact, the average particle size of ACE3/WAT0 was ~7.5 μm, whereas that of ACE0/WAT3 was ~13.2 μm (Figure 3B). Therefore, it is reasonable to consider that the larger particle size contributes low densification of the cement surface in the ACE1/WAT2 and ACE0/WAT3 samples. Moreover, the EDX spectra (inset) of the cement surface layer indicate that P was present after immersion in the artificial saliva. This means that biomimetic apatite re-mineralization may have occurred in the Si-rich granule surface during sparing dissolution of the cements.

Moreover, apatite mineralization helps with the integration of cement with the dentinal canals. As shown in Figure 6, the microstructures with biomimetic mineralized apatite morphology that were about 1–2 μm in diameter were deposited in the dentinal wall. Moreover, apatite was also observed in the dentinal tubules after 5 days of immersion in artificial saliva, implying good in vitro re-mineralization ability (Figure 6). In fact, the EDX mapping analysis also confirmed that the Si-rich zone, which was composed of silicate-based cements, and the elemental variation profiles showed changes in chemical composition after the root canal filling (Figure 6, inset).

### 3.6. Endodontic Space Sealing Ex Vivo of C2Si-Based Pastes

Ex vivo apical microleakage (sealing ability) was evaluated by dye penetration (Figure 7). After placing the single-rooted crack-free human tooth specimens with and without filling paste into rhodamine B dye, the dye penetration occurred in a linear pattern between the filling cements and the dentinal walls. The longitudinal dye penetration occurred obviously in the ACE3/WAT0 and ACE0/WAT3 groups, but there was no significant microleakage in the other two groups.

### 3.7. Antibacterial Activity Evaluation

Figure 8 shows that the different dilute supernatants (from 1 to 1/8) could inhibit the growth of *S. aureus* and *P. aeruginosa*. In particular, there was a similar antimicrobial effect between *S. aureus* and *P. aeruginosa* in the original ion-releasing supernatants, suggesting the C2Si/ATT2 cement may produce appreciable antibacterial efficacy.

## 4. Discussion

The modern concept of dentistry has changed in the last few decades, and dentists continue to strive to conserve more and more teeth. This change in trends has been due to the realization of the fact that natural teeth function more efficiently than any artificial replacement in dental arches. A variety of dental materials has been used, including glass ionomer cement and mineral trioxide aggregate. However, none has met all the criteria set up in the literature [10,11]. In fact, there are numerous situations where the complete physical sealing of the root canal alone does not achieve the aim of optimal treatment. In such cases, multiple functional filling materials with high antibacterial properties, excellent bioactivity (i.e., re-mineralization), and fast setting reaction are more reasonable alternatives [2].

There is wide agreement that the hydration reaction and self-curing rate of pure β-C2Si is very slow. However, the setting time and mechanical evolution of an injectable paste during the self-curing process are two critical factors in root canal filling treatment. Moreover, setting time is also affected directly by many factors, including particle size, powder composition, solution condition, and L/S ratio [23,24,25,26]. Indeed, the smaller the particle size of Ca-silicate powder is, the faster the hydration reaction of the paste will be, and as a result, the low-water paste may quickly produce a hardened cement [27]. The underlying hydration mechanism of C2Si is a constantly evolving field, with no universally agreed-upon mechanism set in root canal fillers. However, the rate of hydration is commonly characterized into four to five stages against time. When C2Si comes into contact with water, a rapid dissolution of ions from the surface takes place, and C2Si hydrolyses into Ca^2+^, OH^−^, and H_2_SiO_4_^2−^. This reaches the maximum hydration rate when the concentration of calcium and silicate ions supersaturate the solution, and the hydration enters the next stage, coined the induction period. Therefore, the initial particle size, specific surface area, crystallization degree, and surface defects of primary powders can directly influence the initial setting process, and mechanical pretreatment such as mechanical alloying [28,29] may increase surface defects and surface reactivity. In our preliminary study, it is evident that a minor amount of Mg–Al–silicate nanofiber (i.e., ATT) is beneficial for self-curing and structural ability, but high ATT contents are detrimental to the self-curing process of C2Si. The presence of ATT in the C2Si cement matrix may readily contribute to its structural densification, and thus the compressive resistance of the self-cured cements can be improved significantly. These phenomena can be explained as follows: low ATT content may anchor the adjacent particles or aggregates in C2Si paste, which may be helpful for reducing the capillary pores and accelerate the densification of cements. On the contrary, if the ATT component is increased highly enough, it may affect the aggregation of the hydrated C2Si aggregates and even result in larger micropores in the cements. Thus, the unexpected setting time and mechanical strength are produced for the composite pastes with high ATT content (≥16%). Based on these primary results, C2Si/ATT2 have been used for the following systematical studies.

In general, low solubility or high stability are prerequisite characteristics of root canal paste, which are of particular importance and necessary for long-term clinical outcomes [30]. Ca-silicate hydrate is very stable in humid environments because the silicate groups readily react with water and then form high poly-condensation silica networks. Therefore, silicate-based cements show high stability under natural conditions unless suffering from strong acid erosion. Indeed, a hydration reaction would readily occur once the C2Si particles make contact with the water-containing medium. Previous researchers tried to avoid this phenomenon (contacting water before preparing paste) when studying C2Si-containing materials. In contrast, we aimed to investigate the influence of ball milling conditions on the potential of physiochemical properties for root canal filling applications. Thus, mixtures with different acetone/water ratios (from 3:0 to 0:3) were used as the ball milling media for as-calcined C2Si powders. Acetone is more volatile than ethanol. Thus, it is reasonable to postulate that the hydration reaction of C2Si powder could be readily terminated once the filling treatment with the acetone–water mixture medium has been conducted. Our results indicate that the powders milled in pure acetone also produce some degree of retarded self-curing reaction, and the initial/final setting time is longer than with those milled in the lower polar mixing media. These primary results suggest that the ball milling medium may influence the particle size of the C2Si powders, and moreover, the partially hydrated superfine C2Si powders may produce an accelerated self-curing response.

As for the mechanical strength evolution of the C2Si/ATT2 cement, on the other hand, it is accepted that when the β phase of C2Si powders react with water, an amorphous, nanoporous C-S-H gel may deposit on the original C2Si particles. As time proceeds, the C-S-H gels may harden and form a solid network, which may be associated with the densification and increase of mechanical strength. It is inevitable that increases of L/S result in more micropores in the gel–solid network, and such microstructures would develop cements with low compressive resistance with a prolongation of time. Evidently, the superfine C2Si particles hydrate more rapidly and possess more compact cement architectures because the growth of C-S-H gel is able to fill the microscopic pore networks and form a denser internal structure. Thus, it is reasonable to assume that the higher the pre-hydration of the C2Si particles is, the slower the migration of the water through the gel layer on the surface layer of the particles or aggregates will be, so that the high-water ball milling medium might be unfavorable for the initial self-hardening and strength development in hydraulic C2Si-based cements.

It is worth mentioning that β-C2Si-containing pastes have been studied by our team and others in endodontic therapy as root filling materials, and more clinical studies have shown a high prevalence of healing after using these materials [8,11]. Microleakage prevention is the pre-requirement to realize root canal therapy. One prominent advantage of silicate-based cement is excellent apatite mineralization activity, which may help to integrate with biomaterials on bone tissue [10,31,32,33]. Previous investigation has confirmed that the smaller the particle size of Ca-silicate-based cements is, the higher the specific surface area of the powders will be, and thus this surface layer reaction would contribute to a better hydration reaction [33,34]. These results indicate that the hydration nature and ball milling condition of the C2Si powders result in different surface microstructures of the cements. It is worth mentioning that the in vitro apical microleakage evaluation showed that the longitudinal dye penetration was more obvious in the ACE3/WAT0 and ACE0/WAT3 groups. These results are abnormal but easily understood because the dye penetration more possibly occurred in the slow hydration or non-dense filling condition. A mild hydration and self-curing process is unfavorable for the densification of cement, and moreover the large particles easily lead to negative effects on microporous nature in its internal structure.

Additionally, another objective of this study was to confirm the antibacterial potential of C2Si-based cement. Till now, bacterial infection after root canal treatment has been the main complication for conventional silicate-based cement [35]. Thus, choosing a filling material with high antibacterial activity helps to decrease or limit the growth of any remaining bacteria, even after treatment in infected defects [36,37,38,39]. Evidently, the antibacterial activity of 2% ATT-containing C2Si-based cement was carried out with *S. aureus* and *P. aeruginosa* using the ion releasing suspension. Such an inhibition level also reflected the expected antibacterial activity of the alkaline C2Si-based cement.

Finally, it should be emphasized that the fast self-curing ability, low solubility, excellent re-mineralization, and even the potent antibacterial potential are several prerequisite characteristics of the root canal filler, which are of particular importance and are necessary for long-term clinical outcomes. In general, the strategies of performance enhancement of silicate-based fillers in most studies, including ours in the present study, are based on the improvement of shortcomings of the pastes with secondary phases and liquid medium conditions. Although it is a very effective method to add other additives or hydration accelerators in the liquid medium, these methods are suboptimal and do not improve the slow hydration nature of the Ca-silicates, especially in the water-deficient root canal environment. In particular, some Ca-silicate pastes have been developed as premixed cements for root canal filling [7,40,41]. In this respect, the partial pre-hydration of Ca-silicate in the dispersant (polar organic liquid) may decrease the requirement for water and even accelerate the self-hardening process. Consequently, it is highly valuable to study the effect of different pre-hydration degrees on the self-curing properties of pre-mixed Ca-silicate-based pastes. As mentioned above, in fact, our studies demonstrate that adjustment of the ball milling medium conditions could be an effective approach to enhance the hydration and self-curing processes, and thus this facile strategy is potentially favorable for other hydraulic silicate-based cement fillers for root canal filling applications. Nevertheless, whether the different pre-hydration degrees in the surface layer of superfine particles influences other self-curing types, such as sulfate or aluminate, to contribute to setting acceleration awaits further investigation. It would also be important to test whether the addition of one-dimensional nanofibers (such as ATT) would promote self-curing paste and thus be a potential structural reinforcement method for stability and even anti-washout.

## 5. Conclusions

In summary, this study indicates that ATT nanofiber additives can influence the setting time, mechanical properties, and long-term stability of C2Si-based cements. It was confirmed that the presence of minor ATT nanofibers (~2%) in C2Si cement did not affect setting efficiency but significantly enhanced compressive strength (over 2-fold) and retarded bio-dissolution due to the contribution of good structural densification. Our studies have also demonstrated that the ball milling medium, especially the presence of an appropriate amount of water (~33 vol%) in the acetone medium, can produce some degree of pre-hydration of the superfine Ca-silicate powder, and thus the self-setting time and mechanical response could both be improved significantly. Additionally, the C2Si/ATT2 composites have been demonstrated to exhibit excellent anti-microleakage and antimicrobial activity. It is believed that such endodontic biomaterial based on composites composed of superfine Ca-silicate and silicate nanofibers is more promising for endodontic treatment.

## Figures and Tables

**Figure 1 materials-15-05043-f001:**
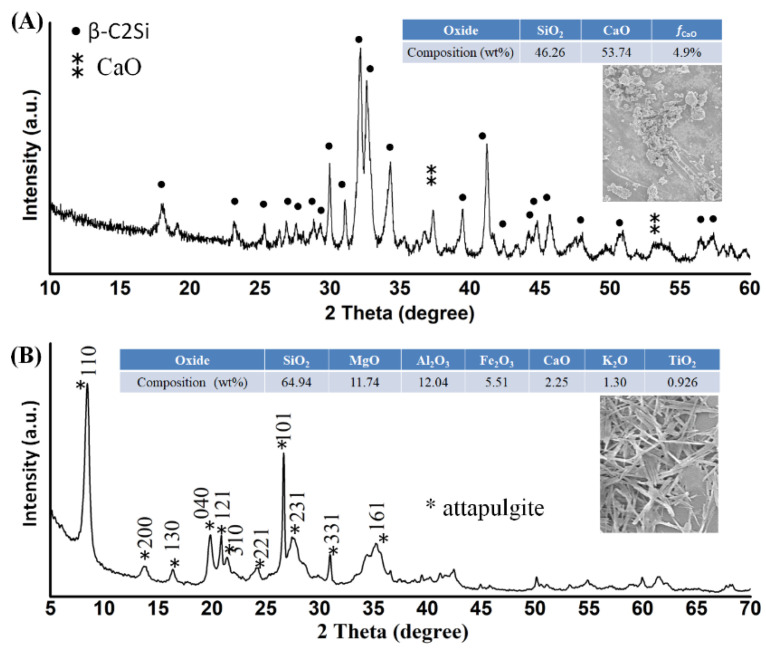
XRD patterns of β-C2Si powders (**A**) and ATT powders (**B**).

**Figure 2 materials-15-05043-f002:**
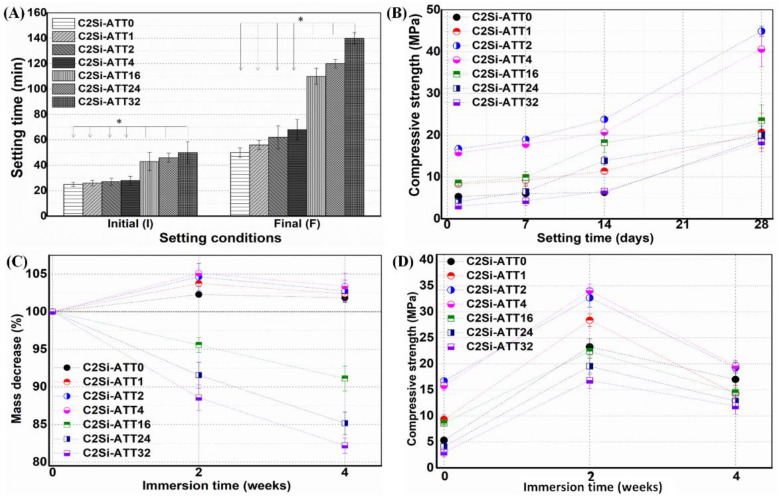
Primary characterization of C2Si/ATT*x* self-curing properties and stability for a long time stage. (**A**) Setting times (I,F) of the pastes; (**B**) evolution of compressive strength; (**C**) mass decay in simulated saliva; (**D**) changes in compressive strength in simulated saliva. * *p* < 0.05.

**Figure 3 materials-15-05043-f003:**
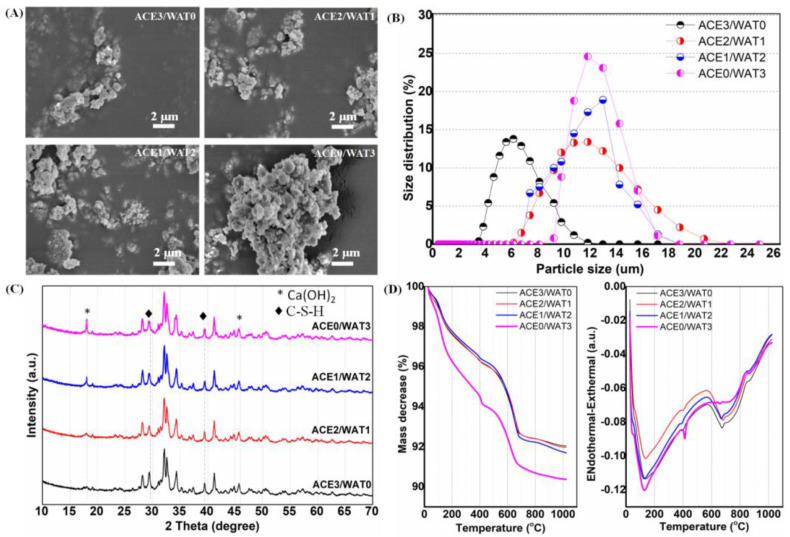
Characterization of the C2Si composite powders milled in the media with different acetone/water ratios for 4 h, respectively. (**A**) SEM observation; (**B**) particle size distribution measurement; (**C**) phasic composition analysis of XRD patterns; (**D**) TG-DTA analysis.

**Figure 4 materials-15-05043-f004:**
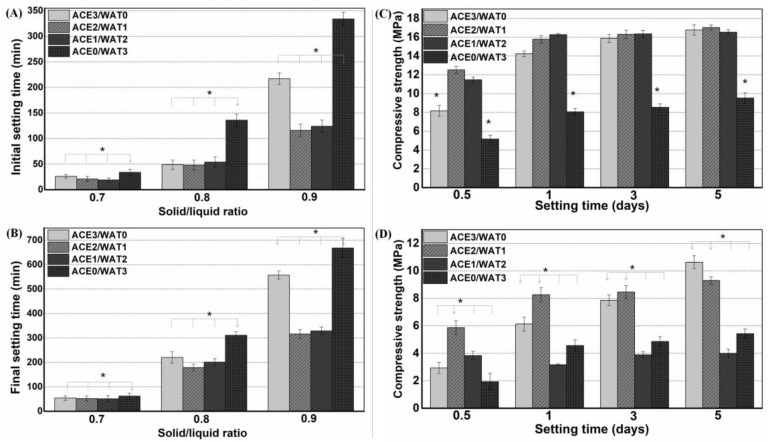
Initial setting time (**A**) and final setting time (**B**) of the C2Si/ATT2 cements with different balling milling conditions, and evolution of mechanical strength of the composite cements with L/S of 0.7 (**C**) and 0.9 (**D**), respectively. * *p* < 0.05.

**Figure 5 materials-15-05043-f005:**
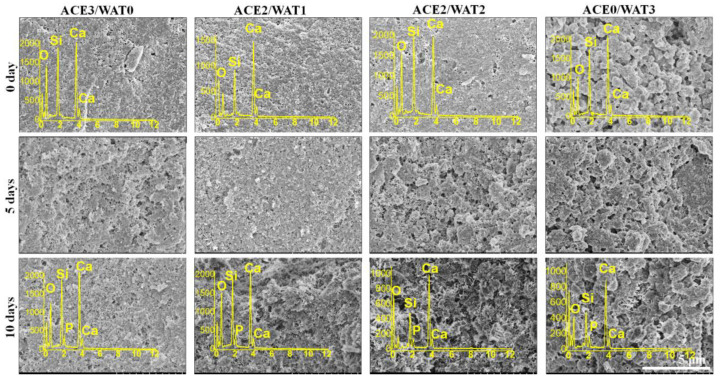
SEM observation and face-scanning EDX spectra (insets) of the C2Si/ATT2 composite cements after immersion in simulated saliva for different time stages.

**Figure 6 materials-15-05043-f006:**
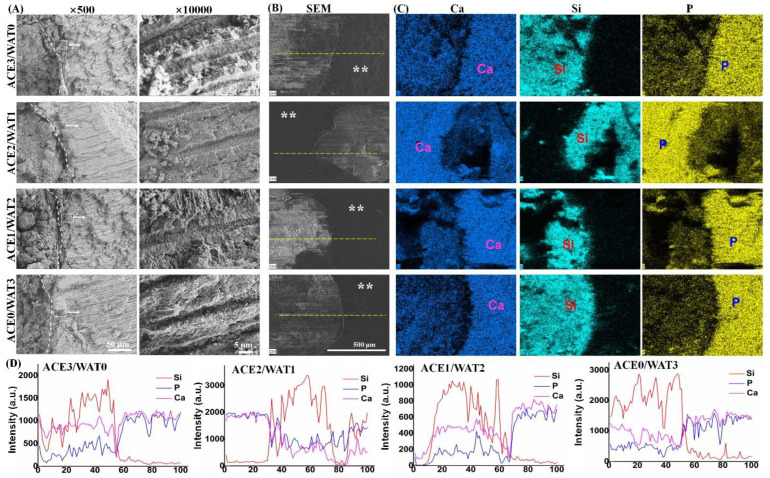
SEM-EDX analysis of the interface of filler/root canal system after soaking in saliva for 120 h. (**A**) SEM mages of the cross-sectional microstructures; (**B**) low-magnification SEM images of fracture surface area between the root canal and cement; (**C**) face-scanning EDX mapping; (**D**) line-scanning EDS profiles along the interfaces zone as shown in (**B**). Dotted-line in (**A**) shows the interface between the apex root of a tooth filled with the C2Si/ATT2 cements. ** dentine; dotted line representing the interface between dentine and paste cement.

**Figure 7 materials-15-05043-f007:**
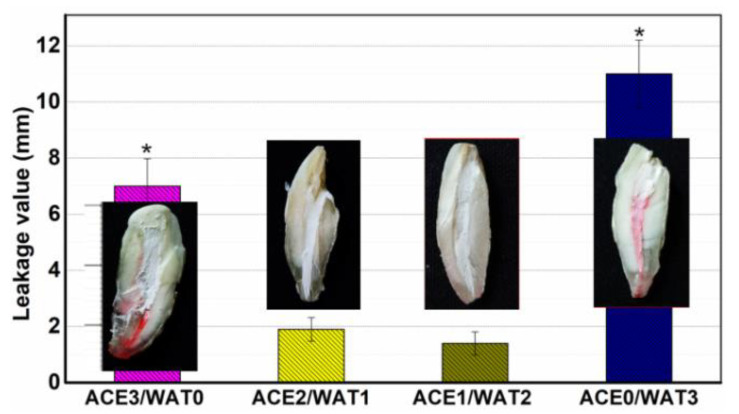
Dye leakage level (average value) of the C2Si/ATT2 cement fillers for 3 days. Inset showing the representative dye penetration of the fracture surface of the teeth. * *p* < 0.05.

**Figure 8 materials-15-05043-f008:**
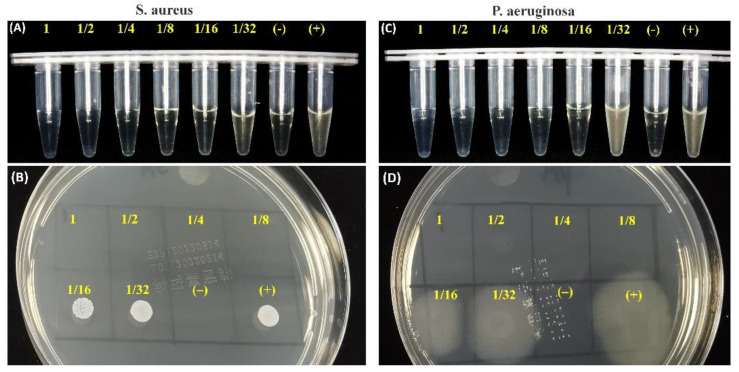
The relative antimicrobial activity against *S. aureus* (**A**,**C**) and *P. aeruginosa* (**B**,**D**) of the different diluted ion-releasing suspensions of C2Si/ATT2 cement at 8 h.

## Data Availability

Not applicable.

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
