# Peer review of "Ball Milling Medium May Tune the Self-Curing Property and Root Canal Microleakage of β-Dicalcium Silicate-Based Cement"

_materials, 2022, doi:10.3390/ma15145043_

Round 1
Reviewer 1 Report
1. Line 156: Missing ethics approval number or certificate number.
2. Line 166: Re sentence "the teeth were filled no paste" suggests to change to "no paste was used"
3. Figure 2D: Spelling mistake on immersion
4. Line 242: What do you mean by appreciable strength?
5. Section 3.3: Refer to which C2Si composition? Abbreviation of ACT should be ACE
6. Figure 7D: EDS profile is too small
7. Section 3.6: The result shown as agar diffusion test whereby the antibacterial zone inhibition was measured. However the result was not clearly shown, moreover the figures of 8A & 8C were not relevant. Does 1-1/8 dilute supernatants show bactericidal?
8. Missing detail results on endodontic space sealing ex vivo (section 2.6) or refer as anti -microleakage in section 3.5 (need to consistence with the term used too). Where are the result of dye penetration in all 50 teeth?
9. Line 372: "In contrat?" or typo error for "In contrast?"
10. Line 399: "and more and more clinical studies" change to "more clinical studies"
Author Response
Reviewer 1
- Line 156: Missing ethics approval number or certificate number.
Authors:Thanks very much for your reminding. This number (I2022521) has been added in the revised manuscript.
- Line 166: Re sentence "the teeth were filled no paste" suggests to change to "no paste was used"
Authors:Thanks for your suggestion. This sentence has been revised as “For the positive control, no paste was used and the surface was coated using nail varnish” base on your comment.
- Figure 2D: Spelling mistake on immersion
Authors:Thank you for your reminding. This error has been revised in the revised manuscript.
- Line 242: What do you mean by appreciable strength?
Authors:This sentence has been revised as “Indeed, the C2Si/ATT2 and C2Si/ATT4 samples showed much higher strength than the other samples.”.
- Section 3.3: Refer to which C2Si composition? Abbreviation of ACT should be ACE
Authors:This spelling error has been revised thoroughly according to your reminding.
- Figure 7D: EDS profile is too small.
Authors:The new Figure 6 (Figure 7 in the primary version) has been re-organized based on your comment.
- Section 3.6: The result shown as agar diffusion test whereby the antibacterial zone inhibition was measured. However, the result was not clearly shown, moreover the figures of 8A & 8C were not relevant. Does 1-1/8 dilute supernatants show bactericidal?
Authors:The contrast of images in Figure 8(A,C) has been adjusted for clearly distinguishing the antibacterial potentials of different cement-diluted dosage conditions. The diluted supernatants and antibacterial zone tests both confirmed the appreciable antibacterial potential of the 1/8 dilute condition.
- Missing detail results on endodontic space sealing ex vivo (section 2.6) or refer as anti -microleakage in section 3.5 (need to consistence with the term used too). Where are the result of dye penetration in all 50 teeth?
Authors:Your comment is very good, and the constructive suggestion is helpful for us to improve the expression of the Result section. We have re-organized the new section 2.6, section 3.5 and section 3.6. The average value of maximal depth of dye penetration was calculated based on the 10 teeth in each group.
- Line 372: "In contrat?" or typo error for "In contrast?"
Authors:This spelling error has been revised based on your reminding.
- Line 399: "and more and more clinical studies" change to "more clinical studies"
Authors:Thanks for your suggestion. This expression has been re-edited in the revised manuscript.
Reviewer 2 Report
The work deals with the effect of attapulgite addition as a filler and ball milling medium (by altering acetone:water ratio) on curing properties of conventional beta dicalcium silicate based dental cements. Furthermore, authors investigated its mechanical properties, sealing performance, antibacterial effect, physical and chemical properties. The manuscript is well designed with experimental data supported with results. It can be published after a minor revision.
C1) The conclusion part is missing the main contents. Authors only mentioned that the addition of ATT amount of approx 2% provides good densification and mechanical property. This sentence needs to be re-defined with the solid values from the results. In this part, authors need to define which sample was the promising one regarding self-curing properties, mechanical properties with which value. Furthermore, which ball milling medium was the best for the powder needs to be defined, instead of “presence of appropriate amount of water in the lower-polar medium”. Please define them according to the samples which were investigated within the manuscript.
Author Response
Reviewer 2
The work deals with the effect of attapulgite addition as a filler and ball milling medium (by altering acetone: water ratio) on curing properties of conventional beta dicalcium silicate based dental cements. Furthermore, authors investigated its mechanical properties, sealing performance, antibacterial effect, physical and chemical properties. The manuscript is well designed with experimental data supported with results. It can be published after a minor revision.
C1) The conclusion part is missing the main contents. Authors only mentioned that the addition of ATT amount of approx 2% provides good densification and mechanical property. This sentence needs to be re-defined with the solid values from the results. In this part, authors need to define which sample was the promising one regarding self-curing properties, mechanical properties with which value. Furthermore, which ball milling medium was the best for the powder needs to be defined, instead of “presence of appropriate amount of water in the lower-polar medium”. Please define them according to the samples which were investigated within the manuscript.
Authors:Thanks very much for your reminding and suggestion. These sentences have been revised as “In summary, this study indicates that ATT nanofiber additives could adjust the setting time, mechanical properties and long-term stability of C2Si-based cements. It is confirmed that presence of minor ATT nanofibers (~2%) into C2Si cement could not affect setting efficiency but significantly enhance compressive strength (over 2 -fold) and retard bio-dissolution due to the contribution on good structural densification. Our studies have also demonstrated that the ball milling medium, especially presence of appropriate amount of water (~33 vol%) in the acetone medium, could produce some degree of pre-hydration of the superfine Ca-silicate powder, and thus the self-setting time and mechanical response could both be improved significantly.”
Reviewer 3 Report
1. Authors must revise the manuscript for grammatical and typo errors.
2. Authors must write a paragraph about mechanical alloying in the introduction part and recommended to add below papers
https://doi.org/10.1016/j.powtec.2015.03.007, https://doi.org/10.1016/j.matchar.2014.11.030, Anal. Bioanal. Electrochem., Vol. 10, No. 3, 2018, 349-361.
3. It is recommended to add the ethanol-water mixture in percentage.
4. During calcination at 850°C, how did authors reduced the chances of oxidation. Which atmosphere they performed calcination?
5. As we all know; contamination is a big problem during ball milling. How did author managed to get the pure cement without contamination?
6. In figure 1, both the XRD is having unidentified peaks. Are they because of contamination?
Author Response
Reviewer 3
- Authors must revise the manuscript for grammatical and typo errors.
Authors:Thanks for your reminding. The manuscript has been revised thoroughly according to your suggestion.
- Authors must write a paragraph about mechanical alloying in the introduction part and recommended to add below papers
https://doi.org/10.1016/j.powtec.2015.03.007, https://doi.org/10.1016/j.matchar.2014.11.030, Anal. Bioanal. Electrochem., Vol. 10, No. 3, 2018, 349-361.
Authors:Your comment is very interesting. The former two documents have been added in the reference lists, but the full paper of last one could not be indexed and downloaded. Some discussion involving the mechanical alloying of primary powder was presented in the discussion section.
- It is recommended to add the ethanol-water mixture in percentage.
Authors:Your suggestion is good, and this expression has been added in the revised manuscript.
- During calcination at 850°C, how did authors reduced the chances of oxidation. Which atmosphere they performed calcination?
Authors:It’s a good question. The sol-gel method involving the Ca salt and colloidal SiO2 could control the free CaO in the calcined powder. In fact, the sol-gel synthesis of β-C2S with TEOS (compares with using silica colloid) could more readily produce high free CaO content possibly due to fast hydrolysis and polycondensation reaction of TEOS in strong acidic medium. Therefore, the free CaO is more difficult to control during β-C2S powder synthesis. In contrast, the silica colloid could maintain a mild hydrolysis and polycondensation reaction and thus the free CaO can be easily controlled. Indeed, when β-C2S powder is used for root canal filling, appropriate amount of free CaO could be helpful for enhancing the antimicrobial potential.
- As we all know; contamination is a big problem during ball milling. How did author managed to get the pure cement without contamination?
Authors:It’s a good comment. We have used the hard-wearing ZrO2 balls to ball mill the powder, and this could reduce and even avoid the powder contamination. In fact, the superfine ZrO2 powder is one of the X-ray retardant candidate in root canal paste. In our experiment, the liquid medium were analytic reagent-grade medium, and so that these pretreatment could significantly reduce the contaminants.
- In figure 1, both the XRD is having unidentified peaks. Are they because of contamination?
Authors:The contaminants in C2Si were mainly controlled by the chemical reagents. We have used the analytic reagents including trishydroxymethyl aminomethane (Tris), colloidal SiO2, ammonia (NH3·H2O, ~28 wt.%), calcium nitrate (Ca(NO3)2·4H2O), acetone, and absolute ethanol. Of course, the attapulgite (ATT) nanofibers was obtained from the deep processing of natural mineral resource, and the minor amount of secondary phase is difficult to fully eliminate. In our XRD patterns, the peaks were all attributed to the β-C2S and free CaO.